



# Modelling Gas Exchange and Biomass Production in West African Sahelian and Sudanian Ecological Zones

Jaber Rahimi[1], Expedit Evariste Ago[2,3], Augustine Ayantunde[4], Sina Berger[1,5], Jan Bogaert[3], Klaus Butterbach-Bahl[1,6], Bernard Cappelaere[7], Jérôme Demarty[7], Abdoul Aziz Diouf[8], Ulrike Falk[9], Edwin Haas[1], Pierre Hiernaux[10], David Kraus[1], Olivier Roupsard[11,12,13], Clemens Scheer[1], Amit Kumar Srivastava[14], Torbern Tagesson[15,16], Rüdiger Grote[1]

[1] Karlsruhe Institute of Technology, Institute of Meteorology and Climate Research, Atmospheric Environmental Research (IMK-IFU), Garmisch-Partenkirchen, Germany
[2] Laboratoire d'Ecologie Appliquée, Faculté des Sciences Agronomiques, Université d'Abomey-Calavi, Cotonou, Bénin
[3] Biodiversity and Landscape Unit, Université de Liège Gembloux Agro-Bio Tech, Gembloux, Belgium
[4] International Livestock Research Institute (ILRI), Ouagadougou, Burkina Faso
[5] University of Augsburg, Regional Climate and Hydrology Research Group, Augsburg, Germany.
[6] International Livestock Research Institute (ILRI), Nairobi, Kenya
[7] HydroSciences Montpellier, Univ. Montpellier, IRD, CNRS, Montpellier, France
[8] Centre de Suivi Ecologique (CSE), Dakar, Senegal.
[9] Satellite-based Climate Monitoring, Deutscher Wetterdienst (DWD), Offenbach, Germany
[10] Géosciences Environnement Toulouse (GET), CNRS, IRD, UPS, Toulouse, France
[11] CIRAD, UMR Eco&Sols, BP1386, CP18524, Dakar, Senegal
[12] Eco&Sols, Univ Montpellier, CIRAD, INRAE, IRD, Institut Agro, Montpellier, France
[13] LMI IESOL, Centre IRD-ISRA de Bel Air, BP1386, CP18524, Dakar, Senegal
[14] Institute of Crop Science and Resource Conservation, University of Bonn, Bonn, Germany
[15] Department of Geosciences and Natural Resource Management, University of Copenhagen, Copenhagen, Denmark
[16] Department of Physical Geography and Ecosystem Sciences, Lund University, Lund, Sweden

*Correspondence to*: Rüdiger Grote (ruediger.grote@kit.edu)

**Abstract** West African Sahelian and Sudanian ecosystems are providing essential services to people and also play a significant role within the global carbon cycle. However, climate and land use are dynamically changing and it remains uncertain how these changes will affect the potential of these regions for providing food and fodder resources or the biosphere-atmosphere exchange of $CO_2$. In this study, we investigate the capacity of a process-based biogeochemical model, LandscapeDNDC, to simulate net ecosystem exchange (NEE) and aboveground biomass of typical managed and natural Sahelian and Sudanian savanna ecosystems. We tested the model for various sites with different proportions of trees and grasses, as well as for the most typical arable cropping systems of the region. In order to describe the phenological development with a common parameterization across all ecosystem types, we introduced soil-water availability in addition to temperature as a driver as seasonal soil water-shortage is a common feature for all these systems. The new approach was tested by using a sample of sites (calibration sites) that provided NEE from flux tower observations and leaf area index data from satellite images





(MODIS). For assessing the simulation accuracy, we applied the calibrated model to 42 additional sites (validation sites) across West Africa for which measured aboveground biomass data were available. The model showed a good performance regarding simulated biomass development. Overall, the comparison of simulated and observed biomass at sites with a dominating land cover of crops, grass or trees yielded correlation coefficients of 0.82, 0.94,

and 0.77 and the Root Mean Square Error of 0.15, 0.22, and 0.12 kg m$^{-2}$, respectively. In absolute terms, the model results indicate above-ground carbon stocks up to 1733, 3291, and 5377 kg C ha$^{-1}$ yr$^{-1}$ for agricultural, savanna grasslands, and savanna mixed tree-grassland sites. Carbon stocks as well as exchange rates correlated in particular with the abundance of trees. The simulations indicate higher grass biomass and crop yields under more humid climatic conditions as can be found in the Sudanian savanna region. Our study shows the capability of

LandscapeDNDC to accurately simulate carbon balances in natural and agricultural ecosystems in semi-arid West Africa under a wide range of conditions, so that it might be used to assess the impact of land-use and climate change on the regional biomass productivity.

## 1 Introduction

Land-cover- as well as land-use changes significantly affect water, carbon (C) and energy exchange processes

between the biosphere and the atmosphere and, thus, climate change (Massad et al., 2019; Pielke et al., 2011). Within the larger biomes, savanna or semi-arid grassland systems have been highlighted as particular important as they are on the one hand assumed to store large amounts of C (Elberling et al., 2003; Scholes and Hall, 1996), and on the other hand experience large C exchanges (Grace et al., 2006). Additionally, savannas are vulnerable to climate change, specifically to changing rainfall pattern or increasing fire intensity or frequency (Grossiord et al.,

2017; Livesley et al., 2011). In particular the role of West African savanna systems for global C cycling has attracted increasing attention over the last decade (Quenum et al., 2019; Bocksberger et al., 2016; Sjöström et al., 2011) due to considerable changes in climate, but also land cover, such as an extension of agriculture and intensification of forest logging (Odekunle et al., 2008). These changes may already have and will further affect the C exchange rates between semi-arid West African savanna ecosystems and the atmosphere, which might not

only affect biomass production, but also might threaten biodiversity as well as the livelihood of people (Dimobe et al., 2018; Hartley et al., 2016; Dayamba et al., 2016).

Hence, to better understand impacts of climate and land use change on biosphere-atmosphere interactions across West Africa, it is important to test our understanding of ecosystem C cycling and C exchange with the atmosphere by using current knowledge to set up ecosystem models and to test if these models are able to a) realistically

represent the sensitive responses of semi-arid ecosystems to climatic variation as well as land-use management,





and b) accurately represent C exchange processes as well as c) the distribution of above- and belowground C pools. Several modelling approaches have been used to describe savanna ecosystem processes and semi-arid cropland development during the past decades, including the application or enhancement of models that have been employed formerly for other ecosystem types under differing climatic and edaphic conditions (Boone et al., 2004; Tews and Jeltsch, 2004; Tews et al., 2006; Grote et al., 2009b; Scheiter and Higgins, 2009; Delon et al., 2019).

In a previous study, different terrestrial models have been evaluated against field observations in order to assess their ability to simulate ecosystem C dynamics and the biosphere-atmosphere exchange of savanna type ecosystems (Whitley et al., 2016; Whitley et al., 2017). It has been particularly highlighted that for these semi-arid systems with seasonally strong variations in plant water availability, it is crucial to represent the plant phenological development, which is something that most models fail to achieve (Pitman, 2003). Indeed, the phenology of deciduous plants is usually related to temperature development, which is generally the limiting factor in temperate ecosystems. For tropical (and sometimes also boreal) regions, phenology is often neglected, and it is assumed that vegetation is foliated throughout the year. In savannas, however, the growth of new tissues is often related to the onset of the rainy season, indicating that water availability is a crucial determinant (Kucharik et al., 2006). Models that account for this influence are still rare. However, in only a few models, water supply has been included as a direct impact on budburst that defines the start of growth for grasses in semi-arid regions (Loustau et al., 1992; Ivanov et al., 2008). Also, Yuan et al. (2007) use simulated soil moisture values as a driver to modify the phenological development of the grass layer of semi-arid steppe, while Jolly and Running (2004) defined the onset of budburst for a grass dominated Kalahari ecosystem as the date at which daily precipitation exceeded potential evaporation, and the start of leaf senescence once soil moisture falls below a defined threshold. Nevertheless, while important for grasses, most trees don't seem to be bound to a minimum water content of the upper soil layers for leaf flushing, possibly because they have excess to deeper water reserves (Do et al., 2005).

Thus, the simulation of C balances in semi-arid tropical systems as can be found in West Africa requires a close and consistent link between rainfall, soil, and vegetation processes. Moreover, as many of the ecosystem types being found in this region are composed of a tree/ shrub layer and a layer of ground vegetation (mainly grasses, sometimes also crops), models should be able to distinguish between different vegetation types and their competition on light and other resources. In addition, it needs to be considered that many of the savanna ecosystems are used by pastoralists or mixed crop-livestock farmers for grazing and cropping (Ker, 1995). Thus, also these management options should be represented in a model.

In this study we parameterized, complemented, and evaluated LandscapeDNDC (Haas et al., 2013), a model framework that uses the soil C-, nitrogen- (N-), and water processes derived from the original DNDC (DeNitrification-DeComposition) models (Li et al., 1992; Kraus et al., 2015). The framework can be flexibly



combined with cohort-based ecosystem models (Grote et al., 2011b) as well as crop growth models (Kraus et al., 2016b). The model framework and its predecessors has so far been used and applied to various natural and

managed temperate and tropical ecosystems such as forests, grasslands, or rice paddies (Kraus et al., 2016b), including savanna grasslands (Grote et al., 2009b). However, so far it was never tested for its suitability to simulate C fluxes over a wide range of different managed and natural savanna ecosystems including different vegetation types.

For this analysis, a literature review has been carried out first to find suitable parameters for the physiological

processes of typical grass and tree species in Sahelian and Sudanian savanna ecosystems. Also, thresholds of soil water availability to leaf development and senescence, which have been derived from twelve sites with ample data available for several years, were introduced. Finally, the model was evaluated against data from other sites representative for the region, including eddy flux measurements, satellite data, and in situ biomass measurements for both managed and natural ecosystems.

The objective of this paper is to test the ability of the LandscapeDNDC model in its current form to simulate C fluxes and stocks for various, representative savanna ecosystem types with varying human management activities within the Sahelian and Sudanian regions.

## 2 Material and methods

### 2.1 Study area

West African semi-arid drylands are located between 15°E-16°W and 07°N-19°N (between the Sahara and the Guinean zone), and spread over 11 countries (Benin, Burkina Faso, Côte d'Ivoire, Gambia, Ghana, Guinea, Mali, Niger, Nigeria, Senegal, and Togo; Fig. 1). Following a transect of decreasing precipitation from South to North there is a gradual transition from forest, woodland, savanna woodland, savanna grassland, to semi-desert grassland (Kaptue Tchuente et al., 2010). Along this gradient, the amount of ground cover and the proportion of woody

species (trees, shrubs, and bushes) decreases and the vegetation becomes shorter. This region is classified into two distinct ecological zones, "Sahelian" and "Sudanian", which differ mainly in terms of precipitation amount and dry season length. In the Sahelian ecological zone which extends over 1.3 million km$^2$, the average annual temperatures vary between 25 and 31 °C and the annual precipitation is between 150 and 600 mm. The length of dry season lasts for seven to nine months annually and the monthly precipitation maximum is in August. The

Sudanian zone of approximately 1.7 million km$^2$ extension is cooler (22–29 °C), wetter (600-1200 mm yr$^{-1}$) and





the dry season length is around four to seven months with monthly maximum precipitation also occurring around August (NASA Power climate dataset, https://power.larc.nasa.gov/).

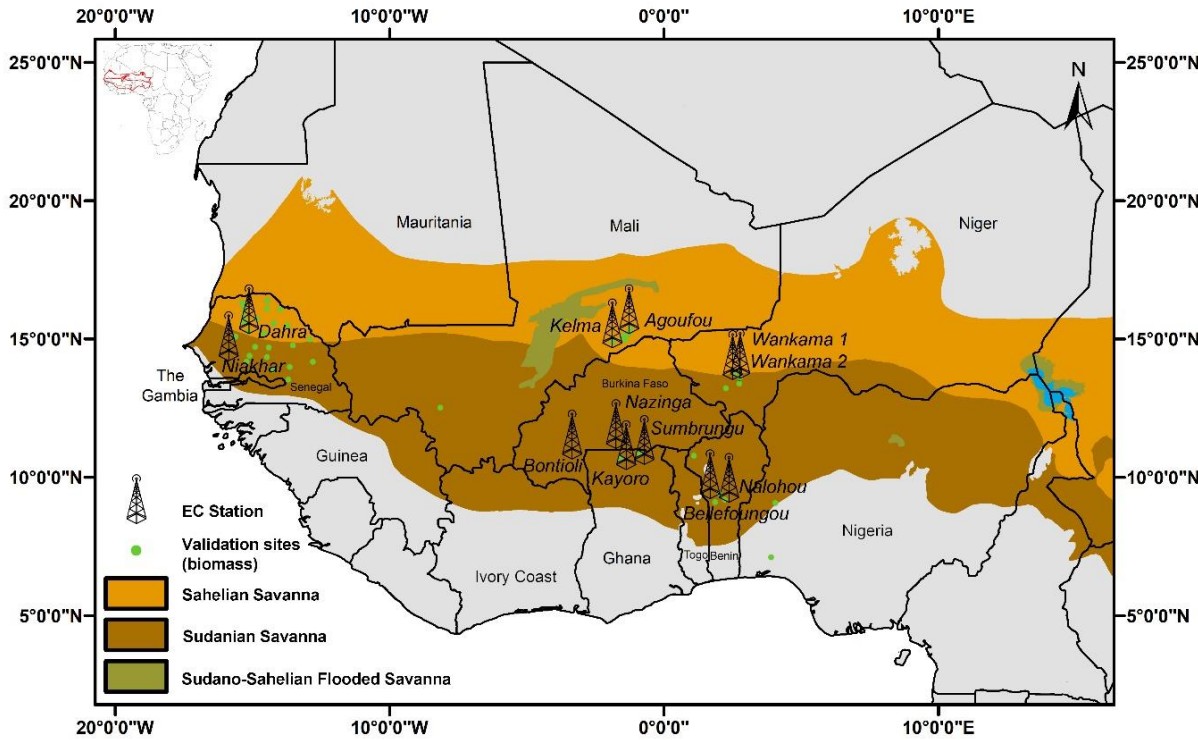

**Figure 1: Map of West Africa showing the Sudanian and Sahelian ecological zones that were derived after Olson et al. (2001). Locations of measurements are indicated as towers (eddy covariance flux stations) or green dots (biomass production).**

## 2.2 Datasets for simulation and evaluation

### 2.2.1 Initialization, climate, and deposition

As model input data, initial soil and vegetation properties as well as daily climate data are required. Land management need to be prescribed as boundary conditions. The soil parameters were bulk density (kg m$^{-3}$), pH, soil texture (i.e. clay, silt, and sand content), organic C and N content (kg kg$^{-1}$), and soil hydrological parameters (i.e. field capacity, wilting point (mm m$^{-3}$)). These were gathered from available literature sources for each site and complemented by information from the Principal Investigators (PI) of these sites. In some few cases, soil



information was complemented with data from ISRIC-WISE (International Soil Reference and Information Centre-World Inventory of Soil Emission Potentials) soil dataset (Batjes, 2009). Similarly, vegetation was initialized with data of amount of grass biomass, tree sizes (average height and breast height diameter), and number of trees ha[-1] as recorded at the sites.

Input climate data were maximum and minimum temperature (°C), precipitation (mm), relative humidity (%), solar radiation (W m[-2]), and wind speed (m s[-1]). These data were either obtained directly from measurements at the sites, or from the NASA-Power climate dataset. With the exception of Kelma, all sites were assumed to have no ground water access down to their maximum rooting depth of 1 m. For the Kelma site, a flooding period was prescribed for the years 2005 to 2007, according to published water content data (Timouk et al., 2009; de Rosnay

et al., 2009). Management was also prescribed for both agricultural sites and savanna grasslands that were occasionally grazed with cattle. Fire impacts were not considered.

In addition to climate, daily input of dry and wet deposition of oxidized and reduced N compounds (nitrate, ammonia, nitric acid, and nitrogen oxide) were provided for the model runs. These were derived from the field measurements within the IDAF (IGAC (International Global Atmospheric Chemistry)/DEBITS (Deposition of

Biogeochemically Important Trace Species)/AFRICA) project, in which also some of our investigation sites were included (Agofou, Banizoumbou, Djougou, and Katibougou). According to this dataset, the total wet deposition of N was estimated to be around 3.2 kg and dry deposition around 3.6 kg N ha[-1] yr[-1] with only little variation between Sahelian and Sudanian ecosystems (Galy-Lacaux et al., 2014).

### 2.2.2 Carbon exchange flux measurements

We collected net ecosystem C exchange (NEE) measurements from 12 flux-tower sites, hereinafter called the core sites, in the Sudanian and Sahelian ecological zones. These data were used for calibrating the new phenological routine of the LandscapeDNDC model and to evaluate simulations for crop-dominated, tree-dominated grassland/woodland, and grass-dominated ecosystems. These eddy covariance measurements, were carried out within the CarboAfrica (www.carboafrica.eu) and AMMA (www.amma-international.org) projects or by the

WASCAL research center (wascal.org).

The spatial distribution and characteristic of the core sites shows that they are well distributed across the major natural and agricultural areas in the Sudanian and Sahelian ecological zones covering grassland savannas (from sparse grasslands to shrublands), woodlands (including seasonally flooded, open, and dense forests), and cultivated land types (major crops: maize (*Zea mays*), millet (*Panicum miliaceum*), sorghum (*Sorghum bicolor*), peanut

(*Arachis hypogaea*), and cassava (*Manihot esculenta*) (Fig. 1; Table 1). Five sites are located in the Sahelian



ecological zone (Dahra-Senegal; Kelma-Mali; Agoufou-Mali; Wankama-1&2-Niger), and seven in the Sudanian ecological zone (Sumbrungu-Ghana; Bontioli-Burkina Faso; Kayoro-Ghana; Nalohou-Benin; Nazinga- Burkina Faso; Niakhar-Senegal; Bellefoungou-Benin). For more information, see Tab. 1 and the references therein.

### 2.2.3 Above-ground biomass- and leaf area measurements

Above-ground herbaceous biomass and crop yield was available from sampled measurements for all core sites except for two of the Sudanian cultivated sites (Kayoro and Nalohou). For the Kayoro site, biomass was assumed to equal that at the Vea site in Ghana, and for the Nalohou site, maize data were taken from the Dassari site in Benin and the Wa site in Ghana, sorghum from the Samanko site in Mali, and cassava from the Ikenne site in Niger, because these sites have a similar climate.

In addition, 42 sites representing crop- (11), grassland savanna- (27), and grassland ecosystems with significant tree contributions (4) from which biomass data are available were used for validating the revised model (Fig. 1). These sites are located in Senegal, Benin, Mali, Ghana, Niger, and Burkina Faso covering overall 70 years of observations within the time period 1984-2018 (not continuous for all sites). 24 sites in Senegal (20 grassland, four savanna woodlands) that were monitored between 1999 and 2017, were provided by the Centre de Suivi Ecologique (CSE) (Diouf et al., 2015). Data from four sites with contributions of trees as well as grasses (Eguerit, Hombori Hondo, Tara, and Timbadior) and one millet site (Bilantao) were obtained from the Gourma region in Mali representing a precipitation gradient from 150 to 400 mm yr$^{-1}$. These field measurements cover the period between 2005-2016, and were gathered within the African Monsoon Multidisciplinary Analysis (AMMA) (Mougin et al., 2019; Mougin et al., 2009). Three more fallow and grassland sites are located in the Fakara region in Western Niger (Banizoumbou, Tigo Tégui, and Kodey). Measurements here began in 1994 taken by the International Livestock Research Institute (ILRI) (Hiernaux and Ayantunde, 2004) and continued until 1999 under the responsibility of the AMMA-CATCH observatory (Cappelaere et al., 2009). Furthermore, we gathered biomass production and yield data covering all major crops of the core sites from three sources. The first dataset containing data for maize, millet, and sorghum for 2004, 2005, 2010, 2015, and 2016 (with different fertilization treatments and different crop sowing windows) consists of 1 site in Benin (Dassari), 2 sites in Ghana (Vea and Wa), as well as sites in Mali (Samanko) and Niger (Sadore) (Faye et al., 2018). The second dataset was obtained in Upper Ouémé Catchment in the Republic of Benin (for 2001, 2002, and 2003) providing information on total above-ground biomass in three sites (Dogue, Beterou, and Wewe) for peanut, maize, and sorghum production with different managements (e.g. different fertilization types/amounts) (Dagbenonbakin, 2005). The third dataset used





for validating LandscapeDNDC were from cassava fields from the sites Ikenne and Oke-Oyi in Nigeria (Sobamowo, 2016).

Leaf Area Index (the half-sided leaf area per unit ground area, LAI) data at 4-day temporal and 500 m spatial resolution from MODIS satellite data (MCD15A3H, https://doi.org/10.5067/modis/mcd15a3h.006), were

downloaded from the Land Processes Distributed Active Archive Center gateway (LP DAAC, https://lpdaac.usgs.gov/) for the pixels where the core sites are located in order to check the seasonality of the vegetation growth simulations.

**2.3 The LandscapeDNDC model**

**2.3.1 Description and parameterization**

LandscapeDNDC is a framework for one-dimensional biogeochemical models, which mainly simulate C, water, and N cycling between the atmosphere, vegetation, and soil at daily to sub-daily temporal resolution for various ecosystems, i.e. arable, grassland, and forest (Haas et al., 2013). We used the version 1.30.4 (ref. 9953) of this model (https://ldndc.imk-ifu.kit.edu/) including the sub-models MeTr$^x$ for soil biogeochemistry and soil respiration (Kraus et al., 2015), ECM for microclimate (Grote et al., 2009a), and the original DNDC routines to

describe the water cycle (Li et al., 1992; Kiese et al., 2011). For grasslands and grass/woodlands the physiological simulation module PSIM (Grote et al., 2011a; Werner et al., 2012) is used, which has been widely applied for forests including sites where ground vegetation needed to be considered (Lindauer et al., 2014; Dirnböck et al., 2020), and also on savanna grasslands (Grote et al., 2009b). However, the model doesn't consider the production of fruits or similar. Therefore, we applied the PlaMo$^x$ module (Kraus et al., 2016a; Liebermann et al., 2020) for

the agricultural plants, which has been developed particularly to described crop growth based on the same physiological processes as PSIM.





**Table 1: Characteristics of eddy covariance sites used in this study (MAT: Mean annual temperature, MAP: Mean annual precipitation).**

| Site Name | Country | Lat, Long | Zone | Ecosystem | Model category | MAT (°C) | MAP (mm) | Soil Type | Period of flux tower meas. | Reference for description |
|---|---|---|---|---|---|---|---|---|---|---|
| Agoufou | Mali | 15.34°N, 01.48°W | Sahelian | Open grassland with sparse shrubs | grass, grazed | 30.2 | 374 | Sand | 2007-2008 | Merbold et al. (2009) |
| Dahra | Senegal | 15.40°N, 15.43°W | Sahelian | Grassland/ Shrubland | grass, grazed | 28.0 | 404 | Sand | 2010-2017 | Tagesson et al. (2015) |
| Kelma | Mali | 15.22°N, 01.57°W | Sahelian | Seasonally flooded open woodland | trees and grass | 29.6 | 650 | Clay-Loam | 2007-2008 | Merbold et al. (2009) |
| Wankama-1 | Niger | 13.65°N, 02.63°E | Sahelian | Fallow land with bushes | grass | 29.5 | 510 | Sand | 2005-2012 | Velluet (2014) |
| Wankama-2 | Niger | 13.64°N, 02.63°E | Sahelian | Cultivated land | agriculture (millet) | 29.5 | 510 | Sand | 2005-2012 | Velluet (2014) |
| Belle-foungou | Benin | 9.79°N, 01.72°E | Sudanian | Protected woodland | trees and grass | 27.0 | 1624 | Loamy-Sand | 2008-2015 | Ago (2016) |
| Bontioli | Burkina Faso | 10.88°N, 03.07°W | Sudanian | Grassland/ Shrubland | grass | 24.9 | 926 | Sandy-Loam | 2005-2006 | Brümmer et al. (2008) |
| Kayoro | Ghana | 10.92°N, 01.32°W | Sudanian | Cultivated land | agriculture (sorghum, peanut, millet) | 28.0 | 994 | Loamy-Sand | 2013-2016 | Quansah et al. (2015), Bliefernicht et al. (2018) |
| Nalohou | Benin | 9.74°N, 01.61°E | Sudanian | Cultivated savanna | agriculture (maize, sorghum, peanut, cassava) | 27.0 | 1190 | Loamy-Sand | 2007-2009 | Ago et al. (2014) |
| Nazinga | Burkina Faso | 11.15°N, 01.58°W | Sudanian | Pristine woody savanna | trees and grass | 27.1 | 961 | Sandy-Loam | 2013-2016 | Quansah et al. (2015), Bliefernicht et al. (2018) |
| Niakhar | Senegal | 14.50°N, 16.45°W | Sudanian | Cultivated land | agriculture (peanut, millet) | 26.9 | 578 | Loamy-Sand | 2018-2019 | Roupsard et al. (2020) |
| Sum-brungu | Ghana | 10.85°N, 00.92°W | Sudanian | Savanna grassland | grass, grazed | 28.3 | 978 | Loamy-Sand | 2013-2016 | Quansah et al. (2015), Bliefernicht et al. (2018) |



While parameters for agricultural crops were mostly species-specifically defined in the model already, information about physiological properties of savanna grass- and tree- species were newly collected from literature. Some properties were assumed ubiquitous for savanna trees and grasses, but we also recognized some species-specific differences between the Sudanian and Sahelian ecosystems. Therefore, we selected *Acacia* (in particular *A. tortilis* also known as *Vachellia tortilis*) as dominant tree genus and *Andropogoneae* (in particular *Andropogon gayanus*)

as dominant grass type for the Sahelian zone. For the Sudanian ecoregion we chose *Burkea africana* as dominant tree species and *Cenchrus biflorus* as dominant grass species (Bocksberger et al., 2016; Geerling, 1985; Sotelo Montes et al., 2014). The parameters describing the specific physiological properties of each species are given in Table 2 for natural and Table 3 for agricultural species.

Both vegetation models (PSIM and PlaMo$^x$) were using the Farquhar approach (with the extension for C4

photosynthesis by Collatz) (Farquhar et al., 1980; Collatz et al., 1992), which required a number of parameters related to enzyme activities (also see Table 2 and 3). Respiration was differentiated into growth respiration and maintenance respiration. Growth respiration was estimated as a fixed fraction of net photosynthesis (25 %). Maintenance respiration was calculated using a linear relationship to N content, but modified by temperature and the relative depletion of C reserves as done by Thornley and Cannell (2000). Carbon was allocated into the different

plant tissues according to Grote (1998), with leaf expansion determined by growing degree sum (see below) and C supplied from previous-year storages.

Evapotranspiration was driven by potential evapotranspiration, calculated with a modified Thornthwaite approach (Camargo et al., 1999; Pereira and Pruitt, 2004; Thornthwaite, 1948) and constrained by soil water availability down to rooting depth (weighted by root mass density) as well as photosynthesis. Without soil-water restrictions,

photosynthesis and stomatal conductance were iteratively calculated based on the Ball-Berry approach (Ball et al., 1987) using a scaling parameter (GSA) and the species-specific maximum conductance as parameters (minimum conductance is generally set to 10 mmol $H_2O$ m$^{-2}$ s$^{-1}$). If relative soil water availability was decreasing below a threshold value, either the stomatal conductance (PSIM) or the photosynthesis itself (PlaMo$^x$) was linearly reduced (Leuning, 1995; Knauer et al., 2015). For the crop-model we assumed this threshold to be when 70 % of the

extractable soil water content (= field capacity – wilting point, see Grote et al. (2009a)) have been depleted. Interception was calculated from LAI, which is the product of specific leaf area and foliage biomass, and specific interception capacity following Gash et al. (1995). For soil water availability, we assumed a rooting soil depth of 1 m for all sites, with no difference between trees and grasses (according to the ISRIC soil data base, available at https://www.isric.org/explore/wise-databases). Fine root abundance was exponentially decreasing with depth, with

roots of grasses slightly more concentrated at the upper soil layers than tree roots (February and Higgins, 2010).





Leaf flushing was assumed to begin when a threshold of cumulative daily temperature sum (growing degree days) had been reached and prolongs throughout a defined period of days. The newly grown foliage was assumed to be fully shed after a period defined by the leaf longevity, and also the shedding needed a defined period of days (Grote, 2007). In order to account for the close relationship between leaf flushing of savanna grasslands and rainfall
events, we included a constraint that foliage starts to grow only if the relative available soil water content was above a threshold value. Similarly, we also included a restriction so that drought-related senescence can only occur after foliage had been at least 90 % developed. All phenological parameters have been defined as the best fit to all Sahelian and Sudanian core sites for natural as well as agricultural systems, respectively (Table 4 and 5).

### 2.3.2 Model setup and initialization

Agricultural sites:

For Kayoro, peanut was planted first each year (June-July), followed by pearl millet or sorghum and the harvest dates were set to mid to end of October. For the Niakhar site, in 2018, pearl millet was sown on May 15th (sowing density = 0.8 m$^{-2}$) and harvested on October 12th. For 2019, peanut was planted at July 05th (sowing density = 8.33 m$^{-2}$) and harvested at end of November. Wankama-2 had been under continuous millet cultivation generally
being planted after the start of the wet season (around 10th of July) (sowing density of around 10,000 pockets (handfuls) per hectare) and harvested shortly after the end of the wet season (mid-October). At the Nalohou site, the landscape was more heterogeneous and the eddy flux measurement was influenced by both mixed crops and fallow bush at the north of the site. Therefore, for this site all crops were planted/grown during the wet season (between April and September-October). The results were post-processed according to the footprint analysis
(which indicates the contribution of each vegetation type to the measured flux for wet and dry season), percentage of area planted by each crop, and cropping pattern in the site footprint area provided by Ago et al. (2014). The simulations used a three-year spin-up with deploying identical management to reach equilibrium. In all simulations, no irrigation and fertilization were applied and tilling always occurred before the planting date, meaning that all straw and grasses that remained on the field were transferred to the soil and incorporated into the
soil C and N pools.





**Table 2: Plant parameters affecting the C and water exchange in the PSIM vegetation model used for simulating biomass growth of natural savanna ecosystem types.**

| Variable name | Description | Savanna trees | | Savanna grasses (C4) | |
|---|---|---|---|---|---|
| | | Sahelian (*A.tortilis*) | Sudanian (*B.africana*) | Sahelian (*C.biflorus*) | Sudanian (*A.gayanus*) |
| Parameters related to C gas exchange | | | | | |
| AEJM | activation energy for electron transport rate (J mol[-1]) | 79500[1] | | 77900[2] | |
| AEKC | activation energy for Michaelis-Menten constant for $CO_2$ (J mol[-1]) | 79430[1] | | 64200[3] | |
| AEKO | activation energy for Michaelis-Menten constant for $O_2$ (J mol[-1]) | 36380[1] | | 10500[3] | |
| AERD | activation energy for dark respiration (J mol[-1]) | 46390[1] | | 50967[4] | |
| AEVC | activation energy for Rubisco carboxylation (J mol[-1]) | 65330[1] | | 78000[3] | |
| AEVO | activation energy for Rubisco oxygenation (J mol[-1]) | 37530[5] | | 55300[3] | |
| HDJ | deactivation energy (J mol[-1]) | 201950[1] | | 210000[3] | |
| SDJ | entropy term (kJ K[-1] mol[-1]) | 650[1] | | 645[3] | |
| KC25 | Michaelis-Menten constant for $CO_2$ at 25 ℃ (µmol mol[-1]) | 275[1] | | 650[6] | |
| KO25 | Michaelis-Menten constant for $O_2$ at 25 ℃ (µmol mol[-1]) | 420[1] | | 450[6] | |
| THETA | convexity term for electron transport (-) | 0.9[7] | | 0.7[6] | |
| QJVC | ratio between electron transport rate and carboxylation capacity (-) | 3.65[8] | 2.27[9] | 9.80[8] | 6.60[10] |
| QRD25 | ratio between dark respiration and carboxylation capacity (-) | 0.03[11] | 0.03[11] | 0.026[12] | 0.038[13] |
| VCMAX25 | max. Rubisco activity at 25 ℃ | 49.6[8] | 32.2[9] | 40.7[8] | 23.9[13] |
| Parameters related to water balance | | | | | |
| GSMAX | max. stomatal conductivity (mmol$H_2O$ m[-2] s[-1]) | 125[14/8] | | 230[15] | |
| GSA | change rate of stomatal conductance with assimilation (mmol$H_2O$ µmol$CO_2$[-1]) | 9.5[19] | | 4.0[19] | |
| H2OREF_GS | relative available soil water content below which stomatal conductance is decreased (-) | 0.3[16] | | 0.4[16] | |
| MWFM | specific interception capacity of foliage (mm m[-2]LAI) | 0.05[17] | 0.5[18] | 0.35[17] | |
| SLA | specific leaf area (m[2] kg[-1]) | 10.8[8] | 8.1[20] | 13.1[21] | 10.6[22] |

[1]Chen et al. (2008), [2]Massad et al. (2007), [3]Boyd et al. (2015), [4]Collatz et al. (1992), [5] Martin et al. (2000); [6]Von Caemmerer (2000), [7]Thornley (2002), [8]Sibret (2018), [9]Kgope and Musil (2004), [10]Kim and Verma (1991), [11]Fürstenau Togashi et al. (2018), [12]Sonawane et al. (2017), [13]Feng and Dietze (2013), [14]Ullmann (1989), [15]Simioni et al. (2000), [16]Baldocchi et al. (2004), [17]de Jong and Jetten (2007), [18]Running and Coughlan (1988), [19]Baldocchi and Meyers (1998), [20]Guenther et al. (1996), [21]Chandra and Dubey (2008), [22]Setterfield et al. (2018)






**Table 3: Plant parameters affecting the C and water exchange in the PlaMo^x vegetation model used for simulating biomass growth and yield at the agricultural sites.**

| Variable name | Description | C4 crops | | | C3 crops | |
|---|---|---|---|---|---|---|
| | | Maize (*Z. mays*) | Sorghum (*S. bicolor*) | Millet (*P. miliaceum*) | Peanut (*A. hypogaea*) | Cassava (*M. Esculenta*) |
| Parameters related to C gas exchange | | | | | | |
| AEJM | See table 2 | 77900 [1] | | | 45000 [2] | |
| AEKC | See table 2 | 59430 [3] | | | 59356 [4] | |
| AEKO | See table 2 | 36000 [3] | | | 35948 [4] | |
| AERD | See table 2 | 50967 [5] | | | 66405 [4] | |
| AEVC | See table 2 | 52000 [6] | 36000 [6] | 47000 [6] | 58520 [4] | |
| AEVO | See table 2 | 55300 [9] | | | 37530 [10] | |
| HDJ | See table 2 | 200000 [11] | | | | |
| SDJ | See table 2 | 630 [6] | | | 646 [11] | |
| KC25 | See table 2 | 650 [7] | | | 270 [8] | |
| KO25 | See table 2 | 450 [7] | | | 165 [8] | |
| THETA | See table 2 | 0.7 [7] | | | 0.9 [7] | |
| QJVC | See table 2 | 6.7 [7] | | | 2.0 [12] | 1.7 [13] |
| QRD25 | See table 2 | 0.02 [5] | | | 0.01 [4] | |
| VCMAX25 | See table 2 | 100 [13] | 50 [14] | 50 [15] | 150 [16] | 100 [17] |
| Parameters related to water balance | | | | | | |
| GSMAX | See table 2 | 350 [18] | 340 [19] | 210 [20] | 650 [21] | 200 [20] |
| GSA | See table 2 | 4 [22] | | | 10.4 [23] | |
| MWFM | See table 2 | 0.05 [24] | | | | |

[1] Massad et al. (2007); [2] Groenendijk et al. (2011); [3] Yu et al. (2001); [4] Farquhar et al. (1980); [5] Collatz et al. (1992); [6] Sonawane et al. (2017); [7] Von Caemmerer (2000); [8] Bernacchi et al. (2002); [9] Boyd et al. (2015); [10] Martin et al. (2000); [11] Kattge and Knorr (2007); [12] Ainsworth and Rogers (2007); [13] Caldararu et al. (2017); [14] Vico and Porporato (2008); [15] Kothavala et al. (2005); [16] Pallas and Samish (1974) ; [17] de Souza Nóia Júnior et al. (2020); [18] Gleason et al. (2017); [19] Körner et al. (1979); [20] Da Matta et al. (2001); [21] Vu (2005); [22] Sellers et al. (1996); [23] Baldocchi and Xu (2005); [24] Running and Coughlan (1988)





**Table 4: Plant parameters related to new phenological processes used in the PSIM model for describing dynamics in natural savanna ecosystem types, derived from measurements at the core-sites of the investigation.**

| Variable | Description | Savanna trees | | Savanna grasses | |
|---|---|---|---|---|---|
| | | Sahelian (*A.tortilis*)[1] | Sudanian (*B.africana*)[2] | Sahelian (*C.biflorus*)[3] | Sudanian (*A.gayanus*)[4] |
| DLEAFSHED | leaf longevity (period from leaf flush to total leaf loss) (days) | 380 | 380 | 365 | 320 |
| $GDD_{emerg}$ | threshold of cumulative temperature sum (°C) | 2000 | 1000 | 4000 | 3300 |
| H2OREF_FLUSH | min. relative available soil water content required to start flushing | 0 | 0 | 0.1 | 0.4 |
| H2OREF_SENES | relative available soil water content at which senescence is induced | 0 | 0 | 0.8 | 0.4 |
| MFOLOPT | Optimum foliage biomass (kg m$^{-2}$) | 1.5 | 1.2 | 0.2 | 0.1 |
| NDFLUSH | period of leaf expansion (days) | 150 | 150 | 80 | 80 |
| NDMORTA | period of leaf senescence (days) | 160 | 180 | 80 | 120 |

[1](Mougin et al., 2019; Timouk et al., 2009; Sjöström et al., 2013), [2](Quansah et al., 2015; Ago et al., 2016), [3](Delon et al., 2015; Hiernaux et al., 2009; Timouk et al., 2009; Baup et al., 2007; Tagesson et al., 2016; Boke-Olén et al., 2016; Velluet, 2014), [4](Quansah et al., 2015; Berger et al., 2019)




**Table 5: Plant parameters that describe phenology and ontology/yield in the PlaMo$^x$ vegetation model for agricultural simulations (taken from the WOFOST model v.6.1 http:// github.com/ajwdewit/WOFOST_crop_parameters).**

| Variable name | Description | C4 crops | | | C3 crops | |
|---|---|---|---|---|---|---|
| | | Maize (*Z. mays*) | Sorghum (*S. bicolor*) | Millet (*P. miliaceum*) | Peanut (*A. hypogaea*) | Cassava (*M. Esculenta*) |
| Parameters related to phenology | | | | | | |
| SLA | See table 2 | 9 | 35 | 20 | 25 | 25 |
| GDD$_{emerg}$ | See table 3 | 110 | 70 | 60 | 120 | 10 |
| GDD$_{flow}$ | GDD for flowering | 860 | 670 | 1000 | 720 | 60 |
| GDD$_{grain}$ | GDD for grain filling | 780 | 820 | 1100 | 836 | 70 |
| GDD$_{matur}$ | GDD for maturity | 1560 | 1400 | 1800 | 1672 | 360 |
| GDD$_{basetemp}$ | Base temperature for GDD calculation | 10 | 11 | 10 | 10 | 0 |

Savanna sites:

Grasslands were supposed to be fully covered with either the Sudanian or Sahelian grass type while mixed grass/woodlands were simulated by considering grass and tree species as different cohorts within the same simulation run (Grote et al., 2011b). Therefore, competition effects between the plant groups depended on the abundance of trees which was characterized by a ground coverage of 80 % in Bellefoungou, 72 % in Nazinga (Sudanian), and 25 % in Kelma (Sahel). This has been initialized by first defining the dimension (height and

diameter at 1.3m) of the average tree at the specific site, and calculating the ground coverage according to allometric relations described in Grote et al. (2020) and parametrized with data from literature (Buba, 2013). These calculations do not assume a difference between species allometry. The total number of trees at the site was adjusted in order to reach the measured total coverage. Grasslands are initialized with a total biomass of 1000 kg ha$^{-1}$ at the beginning of the simulations at all sites and adjusted during two-year spin-up years to a value that

accounts for the competition on light and water at the sites.

**2.4 Statistical analysis**

To identify the relationship between the simulated and measured NEE and LAI, Pearson's correlation coefficient (r) analysis was performed. The correlation coefficient measures the strength of the linear relationship between two variables (here between the simulated and measured NEE and LAI during the crop growing period for the

agricultural sites and the entire year for savanna grasslands and savanna mixed tree-grassland sites).



## 3 Results and discussion

### 3.1 Agricultural sites

Fig. 2. shows measurements and simulation results for NEE and LAI of the four core sites that are used for agriculture (Kayoro, Niakhar, Nalohou, and Wankama-2). It should be noted that LAI simulations are only shown
during the crop-growing period and are otherwise assumed as zero.

For Kayoro in northern Ghana, the predominant cropping pattern between 2013 and 2016 is peanut in rotation with pearl millet (2013, 2015) or peanut in rotation with sorghum (2014, 2016). Simulated seasonal dynamics and magnitudes of NEE and LAI are well in accordance with measurements (r= 0.79 and 0.76, respectively; Fig. 2). However, in 2013 the model overestimated LAI and therefore also C uptake during the crop-growing period (-55.5
kg C ha$^{-1}$ d$^{-1}$ as estimated from measurements whereas -72.1 kg C ha$^{-1}$ d$^{-1}$ was simulated). A possible reason is the occurrence of weeds that may have prevented peanut and millet to grow to its full potential; or that the fetch of the eddy covariance tower extended beyond the investigation area were less productive plants or bare land result in a reduction of the average data from measurements (Quansah et al., 2015).

At Niakhar, a representative site for the so-called "groundnut basin" of Senegal, either pearl millet (2018) or peanut
(2019) was grown in an annual rotation. Despite an underestimation of LAI for millet peanut, both by approximately 30 %, the deviations between measured and modeled NEE were relatively small (r = 0.80). A possible uncertainty at this site is the presence of *Faidherbia albida* trees (6.8 trees ha$^{-1}$) that typically show a different phenology which explains a negative NEE (C uptake) during the dry season and also an increase of LAI and productivity during the crop-growing period, which was not considered by the simulations.

At Nalohou, located in the Ara watershed in the northern part of Benin, all four crops were planted simultaneously. NEE and LAI simulations agree well with measurements showing average correlation coefficients of 0.74 and 0.86, respectively. This site is a typical cultivated savanna ecosystem, which means that it is simultaneously covered by crops, herbs and shrub savanna. According to a footprint analysis presented by Ago et al. (2014), crops contribute to 77 % of the C exchange fluxes only. In addition, it is also known that other crops than those
considered in the simulations, i.e. yam (*Dioscorea alata*), had been planted within the footprint area of the tower, increasing the uncertainty. Overall, the long vegetation period of Nalohou caused it to become a substantial C sink of 2814 kg C ha$^{-1}$ yr$^{-1}$.





**Figure 2: (A-D):** Time series of the observed and modeled NEE (kg ha⁻¹, negative values represent a flux into the biosphere, positive values a flux towards the atmosphere) and LAI (m² m⁻²) as well as precipitation, temperature, and soil moisture content (10 cm) at four different crop-dominated sites in Sahelian (D) and Sudanian (A-C) ecological zones.

For Wankama-2, located in the Southwest of the Republic of Niger, annual rotations with pearl millet remained unchanged throughout the 7-year study period (2005-2012). Simulations correlated highly with NEE and LAI with

coefficients of 0.84 and 0.82, respectively. Also, simulated patterns of biomass development were matching observations closely. The simulated C sink was on average 810 kg C ha$^{-1}$ yr$^{-1}$, whereas estimates based on measurements were 720 kg C ha$^{-1}$ yr$^{-1}$. However, the simulations indicate a larger variability of yields than indicated by measurements, leading for example to smaller than average yields during 2009-10 (0.12 kg m$^{-2}$ compared to the average of 0.21 kg m$^{-2}$). These years were somewhat dryer than average (320 mm of annual

precipitation in 2009-10 while the normal amount is around 410 mm yr$^{-1}$), indicating that the model seems to be too sensitive to changes in soil water availability. Other uncertainties are potentially varying planting densities.

The results of the agricultural core sites were obtained by calibrating the parameters that determine the maximum biomass and yield values separately for all five crops using observations. Accordingly, the deviation between measured and simulated biomass/yield values across all sites have been minimized resulting in an overall

correlation coefficient of 0.93 (Fig. 3A). Simulating biomass production for six further sites (17 years) across West Africa with the derived parameters and no further adjustment yielded an overall correlation coefficient of 0.82 (Fig. 3B), indicating that the model was well suitable to represent the development of the major crops throughout the investigated area.

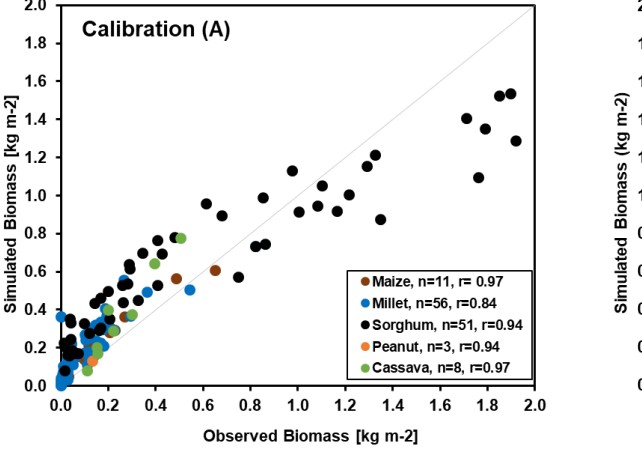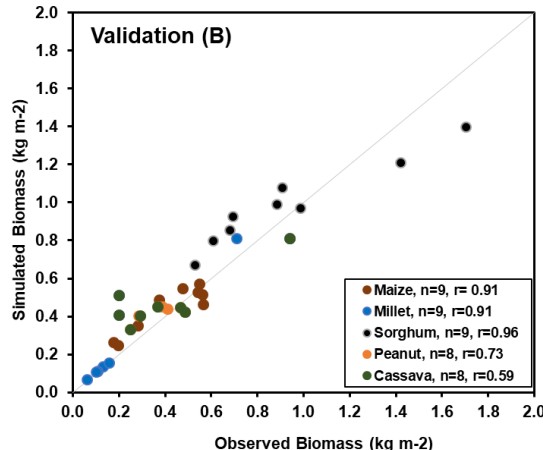

**Figure 3: A) Observed and simulated annual biomass/yield production for crop-dominated sites involved in the calibration process over all 18 investigated years, and B) the same for 17 years at 6 additional validation sites.**



### 3.2 Savanna grasslands

Fig. 4. shows the comparison of measured and simulated NEE and LAI for the five grass-dominant core sites used for calibration. They are differentiated according to their location in Sahelian (A-C: Agoufou, Dahra, and
Wankama-1) and Sudanian regions (D-E: Sumbrungu, Bontioli).

The Agoufou site is located in the southern part of the Gourma region in Mali, which is a typical Sahelian grassland with 2-3 % tree cover (that was neglected in the simulation), at which occasional livestock grazing occurs. Correlation coefficients between the measured and modeled NEE and LAI over the period of 2007-2008 were 0.80 and 0.88, respectively. Some small deviations after a sudden LAI decrease in the middle of the first vegetation
period can be seen (Fig. 4A). This was most likely caused by a minor fire (Samain et al., 2008), but since it is not considered as a driver here, neglecting this event results in an overestimation of LAI as well as NEE for the second half of the period. Below-ground respiration also seems to be overestimated after the end of this vegetation period, which might be caused by the model assuming litter decomposition of grass material that was fully or partly removed by the fire or management activities. Overall, measurements indicate that the Agoufou site acted as a C
sink of 890 kg C ha$^{-1}$ yr$^{-1}$ while the net C sequestration according to simulations is 761 kg C ha$^{-1}$ yr$^{-1}$.

At the Dahra site, which is in the central part of Senegal, the footprint of the eddy-covariance tower is also largely dominated by grass vegetation with occasional trees leading to a negligible tree coverage of 3% (Tagesson et al., 2015). There was a general good agreement with NEE data (r = 0.67), even though the simulations did not capture the peak net C uptake rates during the wet periods. Phenology also started too late, particularly in 2016 (Fig. 4B).
The underestimation could be related to this site being more productive than in general of this region because it is relatively high humidity and nutrient availability (Tagesson et al., 2016). Nevertheless, both measurements and simulations indicate that Dahra was a substantial C sink for all years (2015-2017).

The Wankama-1 site in the Southwest of the Republic of Niger is known as a fallow savanna (herbs and shrubs) where the land-use remained unchanged from 2005 to 2012. Correlation coefficients between measured and
simulated NEE and LAI were 0.77 and 0.78, respectively. However, in some years such as 2006 and 2012, the model assumed longer vegetation periods and thus higher NEE which might be caused by a lack of sensitivity to local drought, unusually intensive grazing, or diseases that have not been adequately considered in the simulations (Fig. 4C). The average simulated NEE for Wankama-1 was an uptake of 1894 kg C ha$^{-1}$ yr$^{-1}$ (the measured value indicates 1505 kg C ha$^{-1}$ yr$^{-1}$ but major data gaps prevent the calculation of a meaningful annual average).






**Figure 4 (A-C): Time series of the observed and modeled Net Ecosystem C Exchange (kg ha⁻¹, negative values represent a flux to the biosphere, positive values a flux to atmosphere) and Leaf Area Index (m² m⁻²) as well as precipitation, temperature, and soil moisture content (10 cm) at different grass-dominated sites in Sahelian sites.**



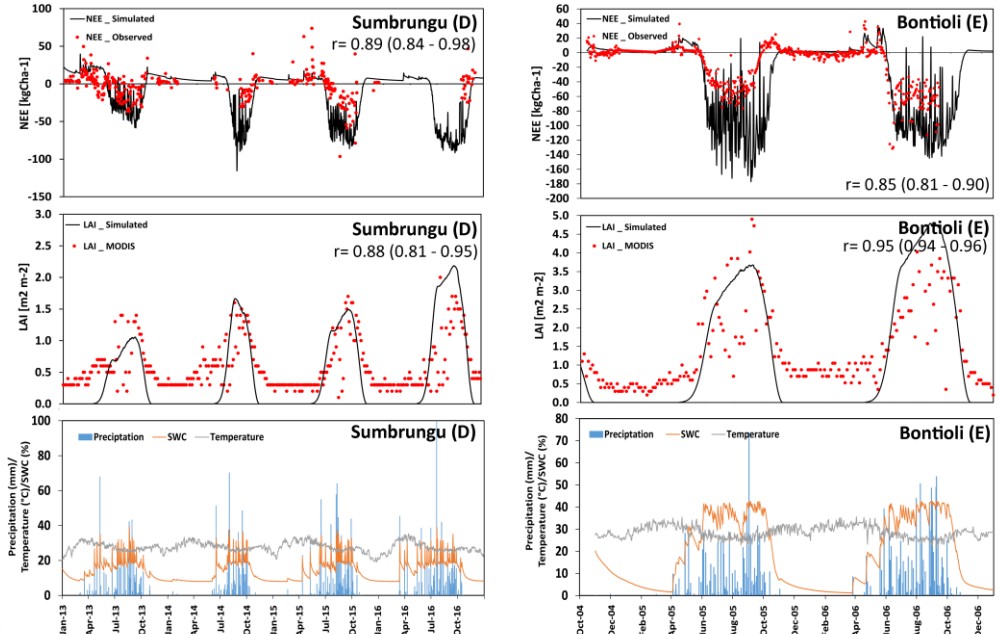


**Figure 4 (D-E): Time series of the observed and modeled Net Ecosystem C Exchange (kg ha⁻¹, negative values represent a flux to the biosphere, positive values a flux to atmosphere) and Leaf Area Index (m² m⁻²) as well as precipitation, temperature, and soil moisture content (10 cm) at different grass-dominated in Sudanian sites.**

Regarding the Sudanian ecozone, the first core site is Sumbrungu, located in Ghana's Upper East Province. The

overall correlation between NEE measurement and model simulations were overall high (r = 0.89), even though

simulations seemed to slightly overestimate the fluxes. According to the simulations, the maximum C uptake was

in August and September at a rate of 29.3 kg C ha⁻¹ d⁻¹. The average C loss over the dry period varies considerably

from 4320 kg C ha⁻¹ (in 2013) to 1487 kg C ha⁻¹ (in 2015), correlating strongly with the annual precipitation (679

mm of annual precipitation in 2013 while the normal precipitation amount is around 978 mm yr⁻¹).

The second Sudanian site is Bontioli in the Southwest of Burkina Faso. Here, NEE was in absolute values again

overestimated by the model during the wet season. Nevertheless, simulations do capture the transition phase from

dry and wet period (Fig. 4D) although the simulated vegetation period was about 2-3 weeks longer than indicated

by the eddy-covariance measurements. However, there was a good correlation between measured and simulated

LAI (R = 0.95), possibly indicating that the efficiency of the plants in capturing C were underestimated. It might

be caused by higher N availability than indicated by the deposition regime, for example due to deposition of fertile

ash from nearby fires (Bauters et al., 2018).



Similar to the core crop sites, we used the parameters obtained for the core sites for the validation of the model's ability to represent grassland savanna sites. In Fig. 5A the above-ground biomass production simulated by the LandscapeDNDC model is compared with field observations of 23 samples from the 5 core sites. It demonstrates

that the simulated above-ground biomass production generally agreed with the observations for all studied sites in Sahelian as well as Sudanian ecological zones (correlation coefficient of 85 %). The validation exercise with additional 27 grass-dominated sites across Sahelian and Sudanian ecological is shown in Fig. 5B. It should be noted that some biomass yields in the validation sites are considerably larger than those found in any of the core-sites.


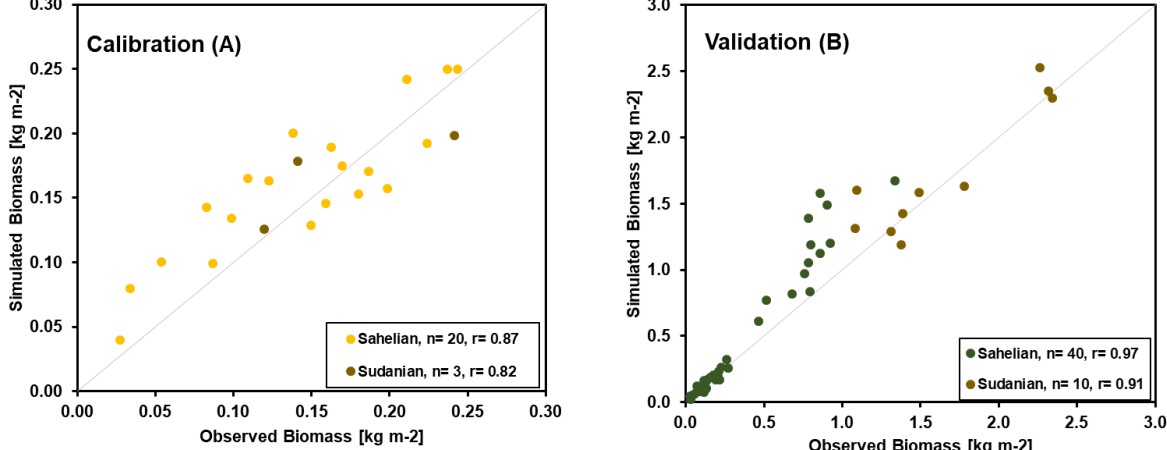

**Figure 5: Correlations between the observed and modeled biomass production at different grass-dominated sites in Sahelian and Sudanian ecological zone for the calibration and validation data.**

### 3.3 Savanna mixed tree-grassland sites

We furthermore investigated the NEE and LAI of three grassland sites with considerable but different tree

contribution. Two of which are located in the Sudanian zone (Nazinga and Bellefoungou) and one in the Sahelian zone (Kelma).

For the Nazinga site in southern Burkina Faso the model was able to simulate the fluctuations in NEE and LAI over time series well (correlation coefficient of 0.79 and 0.85, respectively). However, the model underestimates ecosystem respiration during the dry period (especially in 2013), while LAI values (for both trees and grasses)

were well simulated (Fig. 6A). The high variability of measured fluxes showed C releases during wet periods, indicating that some uncertainty exists with respect to species properties and the measured footprint, possibly involving other species during specific periods (Bliefernicht et al., 2018).





The Kelma site, a facility located in the southern part of the Gourma region in Mali, is specific with respect to its water supply because it is a seasonally flooded open woodland. Therefore, it can be expected that biodiversity is larger and species composition overall diverges from other Sahelian zones. Patterns and magnitudes of NEE and LAI were generally well simulated by the model, although the onset of the vegetation period was estimated somewhat too early by the model (Fig. 6B). Furthermore, measurements indicated high levels of ecosystem respiration occurring at the end of the dry period in 2005, which could not be fully represented by the model. Such rewetting events are assumed to be related to increased decomposition and have been observed before (Epron et al., 2004; Grote et al., 2009b) but remain challenging for soil C models to capture (see Fraser et al. (2016)).

In contrast, LAI at Bellefoungou in the Djougou district located in the northern part of Benin, was overestimated during the peak of the wet season and underestimated during transition phases (Fig. 6C), resulting in a relatively small correlation coefficient between the simulated and measured NEE and LAI (0.63 and 0.52, respectively). Again, this may be related to a relatively high diversity of (tree) species at the site as indicated in literature (Mamadou, 2014) which might result in average ecosystem properties and responses that are different from a simulation with only two species. Nevertheless, the simulated NEE indicated a similar cumulative annual NEE budget of about -5660 kg C ha$^{-1}$ yr$^{-1}$ as compared to measurements.

Since measured foliage or above-ground biomass data of trees are unavailable, we compare measured and simulated herbaceous biomass, as harvested at the peak of the vegetation period only (Fig. 7). For the five sites and across the two climate zones, simulations and measurements were highly correlated (correlation coefficients are 0.95 and 0.64 for the Sahelian and 0.98 and 0.90 for the Sudanian region, for calibration and validation, respectively).





Figure 6 (A-C): Time series of the observed and modeled Net Ecosystem C Exchange (kg ha⁻¹, negative values represent a flux to the biosphere, positive values a flux to atmosphere) and Leaf Area Index (m² m⁻²) as well as precipitation, temperature, and soil moisture content (10 cm) at different tree-dominated grassland/woodland sites in Sahelian (Kelma) and Sudanian (Nazinga, Bellefoungou) ecological zones.





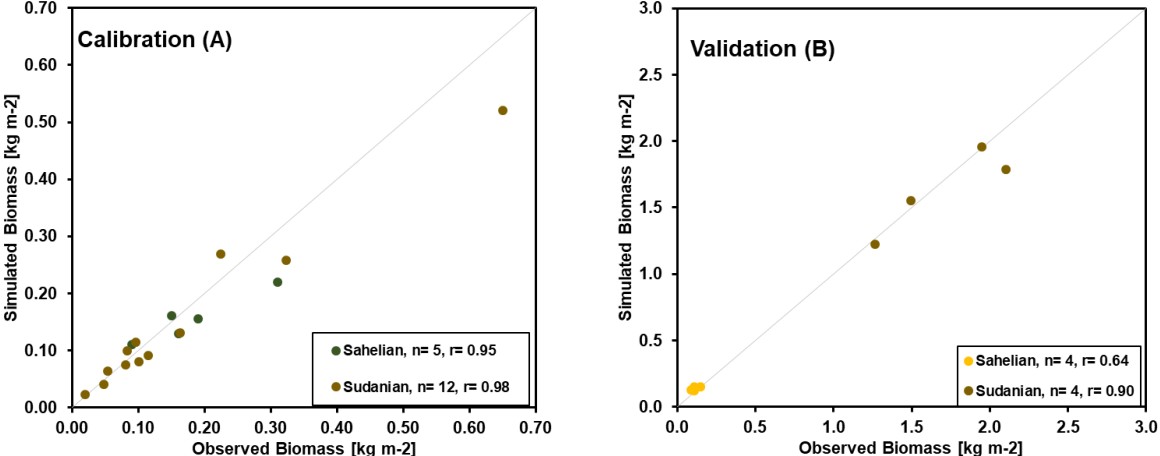

**Figure 7: Correlations between the observed and modeled biomass production at different tree-dominated grassland/woodland sites in Sahelian and Sudanian ecological zone through calibration and validation process.**

## 4 Conclusions

Biogeochemical models can be used to determine terrestrial pool development and fluxes at temporal resolutions and regional scales that cannot be covered by field measurements. However, calibration and evaluation is needed for a range of representative sites in order to be reliable for assessing effects of environmental and anthropogenic (management) changes. Therefore, this study presents the - to date - most extensive calibration and evaluation exercise of a biogeochemical model for natural and agricultural ecosystems in the Sahelian and Sudanian ecological zones of West Africa. More specifically, the LandscapeDNDC model framework has been applied to 54 (12 core sites plus 42 validation sites) intensive investigation sites where eddy-covariance and/or biomass measurements were available for more than one year. These sites were covering the major natural vegetation types as well as the most important agricultural vegetation classes, also considering an appropriate management.

The results show that the parameterized and complemented LandscapeDNDC model is able to represent C fluxes and pools for croplands, pure grasslands as well as tree/grass mixtures in various combinations independent of site quality and climatic conditions. Deviations from measurements could be mostly explained by simplifications in species biodiversity, which, however, did affect overall carbon exchange and yield to only minor degrees. While C-fluxes were mostly well represented by the model, soil C emission following re-wetting of dry soils was occasionally underrepresented, which is a common problem also in other models. Since the respective periods were relatively short and didn't occur at all sites, the overall effect on the annual C balance was rather minor.



Since the model only considers one representative plant species per type in natural ecosystems, it is not able to
account for specific species combinations or a shift in species abundance that might change ecosystem sensitivity
over longer periods. It is, however, feasible to run scenarios of land cover and land use changes that can happen
rather quickly and investigate the impact on the regional carbon cycle considering an altered abundance of different
ecosystems. Also, scenarios of environmental changes (e.g. changes in rainfall patterns, temperature, length of
drought/rainy periods) that can development in relatively short periods of time and thus are faster than ecological
adaptation responses, can well be investigated. With the evaluation presented here, the model is rendered particular
suitable as decision-support tool to explore climate smart agricultural practices, as well as various management
practices in natural and agricultural land use systems in West Africa on biosphere-atmosphere exchange.
Furthermore, since the model is particularly designed to simulate trace gas exchanges (i.e. because of its detailed
consideration of soil processes), also greenhouse gas emissions other than carbon releases can be simulated (e.g.
$N_2O$ and $CH_4$). Thus, it could be applied to assess further management effects such as different fertilization
practices on West African semi-arid ecosystems, and to estimate impacts on greenhouse gas emissions as well as
on plant growth. However, it would be good if further evaluation with respect to such trace gas exchanges could
be provided, particularly covering rewetting events.

*Code Availability*: The LandscapeDNDC model is available online at https://ldndc.imk-ifu.kit.edu/.

*Data Availability*. The EC data are available from the AMMA-CATCH database (bd.amma-catch.org), Fluxnet
(http://fluxnet.ornl.gov), WASCAL Data Discovery Portal (Portalwascal-dataportal.org) and CarboAfrica
(http://www.carboafrica. net/indexen.asp). The NASA POWER climate and the MCD15A3H version 6 MODIS
Level 4 data are available from NASA's Surface Meteorology and Solar Energy Web Portal
(https://power.larc.nasa.gov/) and the Land Processes Distributed Active Archive Center
(https://lpdaac.usgs.gov/), respectively. Data products from this analysis are available on request from the
corresponding author.

*Author contribution*. J.R., R.G, E.H, and K.B.B conceived and designed the study. J.R., R.G and D.K. pre-proceed
the data, parameterized, and validated the model. E.E.A., A.A., S.B., J.B., B.C., A.A.D., U.F., P.H., H.T.T., O.R.,
A.K.S. provided all the flux and biomass measurement data for the analysis. J.R. and R.G wrote the first draft with
all authors contributing to further revisions.

*Competing interests*. The authors declare that they have no conflict of interest.



*Acknowledgement*. The authors are thankful to the UPSCALERS project (AURG II-1-074-2016) which is part of the African Union Research Grants financed through the Financing Agreement between the European Commission and the African Union Commission (DCI-PANAF/2015/307-078). Furthermore, part of the climate data was obtained from the NASA Langley Research Center (LaRC) POWER Project funded through the NASA Earth Science/Applied Science Program. The West African Science Service Center on Climate Change and Adapted Land Use (WASCAL) UPSCALERS project (AURG II-1-074-2016). Finance support for open access publishing was granted from the Library of the Karlsruhe Institute of Technology.

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
