# Peer review of "Modelling Gas Exchange and Biomass Production in West African Sahelian and Sudanian Ecological Zones"

_Geoscientific Model Development, 2020_

## Referee Comment (RC1)

**Comments for gmd-2020-417**

In short, this study intends to evaluate the suitability of the model LandscapeDNDC in capturing the aboveground biomass for different vegetation type (croplands, pure grasslands as well as tree/grass mixtures) in the Sahelian and Sudanian ecological zones of West Africa. This seems the first step for furthering assessing the impact of climate and land use change. The authors conducted extensive calibration and validation. Overall, I am enjoy reading the manuscript. I have some suggestions that might be useful for improving the paper.

1. The authors underscored the importance of the soil water availability to this special region and ecosystem, they considered 70% of the extractable soil water content, I am wondering whether it is vegetation specific since there are three distinct types here (croplands, pure grasslands as well as tree/grass mixtures)? I suggest adding a sensitivity test to illustrate its influence. Or, just set different thresholds for different vegetation (this might be doable since the simulation here was conducted at site level instead of regional level).

2. For me, the LAI at Niakhar was underestimated, and that at Wankama 1 was overestimated, NEE at Bontioli was overestimated, is it possible to have more discussion about the reason or adjust the parameters to have a better capture?

3. There is a long description about the model in 2.3.1, I suggest adding the important equations to make it easy to understand the revenant biophysical process of the model since not all the readers are familiar with the model. For example, how is the actual evaporation calculated from the potential evaporation? Also, Thornthwaite approach mainly depends on temperature, it seems water content is important in this special region with large variation of precipitation, so, how is the impact on the result? How is the performance of the modelled evaporation compared to the flux observation?

4. How is the parameters relevant to soil water content (field capacity and wilting point) at each site? Please clarify.

5. I suggest adding the vegetation distribution in Fig. 1, which is more intuitive and easier to understand.

6. The authors used the Modis LAI, which is 500-m resolution, if the grid is a mixture of different vegetation, it might have a big impact on the validation. Such discussions are needed.

7. Line 318-320, how is the standard for spinup? It says it accounts for the competition on light and water at the sites. Please clarify this in detail.

8. Minor, I think gas exchange might be removed from the title since it is not reflected in the main body.

9. Minor, I fell the abstract is quite long and should be more concise.

10. Minor, superscript in the figure is ignored, please modify.

---

## Referee Comment (RC2)

The authors evaluated the ability of LandscapeDNDC in simulating carbon balance over different ecosystems in West Africa. The topic of this work is important, which may provide a possibility of using this model to assess the impact of land-use and climate change on the regional biomass productivity. I have some comments which may improve the manuscript to a certain extend.

Specific comments:

Lines 203-207: Does the author validate the MODIS LAI data with field measure LAI in the studied sites?

Lines 209: I would put some key functions, such as the calculation of water stress factor, in this section.

Some results were not fully discussed. For example, in Figure 1(A), the LAI was overestimated in 2007 and the NEE was overestimated at the very beginning of the growing season.

Although this manuscript focus on evaluating the ability of LandscapeDNDC in simulating gas exchange and biomass production simulation, the simulation of soil moisture is very important especially in some arid or semi-arid sites (such as Agoufou, Dahara, and Wankama). Do you have the field measured soil moisture? If you have, I suggest the author validate the simulation of soil moisture of LandscapeDNDC.

For the validation of NEE, I would like to partition NEE into gross primary productivity and ecosystem respiration. After that, the gas exchange part can be in-depth discussed.

---

## Author Response (AR1)

**Point-by-point response for:**

gmd-2020-417: 'Modelling Gas Exchange and Biomass Production in West African Sahelian and Sudanian Ecological Zones'

General remarks (both reviewers)

We are grateful for the remarks and suggestion that we happily used to improve the manuscript as described in detail below. In particular, we added a new figure that evaluates satellite derived LAI information (Fig.2), complemented Figure 1 and Table 1 with additional site information, improved the model description including new parameter indications, and added information into Figures 3, 5, and 7 for a better evaluate the representation of water balance simulations. Along the revision, a couple of minor corrections have been made regarding language or logical problems. As far as these led to additions to the text, they are all indicated in track-change mode as are the changes that had been done in response to the specific remarks.

Specific answers to the remarks (R: Repeated remark of the reviewer; A: Answer, **concrete changes in the text as indicated with bold letters**)

Reviewer 1

R1: The authors underscored the importance of the soil water availability to this special region and ecosystem, they considered 70% of the extractable soil water content, I am wondering whether it is vegetation specific since there are three distinct types here (croplands, pure grasslands as well as tree/grass mixtures)? I suggest adding a sensitivity test to illustrate its influence. Or, just set different thresholds for different vegetation (this might be doable since the simulation here was conducted at site level instead of regional level).

A1: The extractable soil water has in fact been set from field measurements per site as the difference between field capacity and soil wilting point, stratified in different layers. The 70% of extractable water has been set as the threshold value after which photosynthesis is linear reduced until it is zero when the wilting point is reached. **For consistency reasons we thus added the respective parameter H2OREF_A into Table 3 and refer to it in the text. However, we found in the FAO database that 70% is considered rather at the upper end of the range and a value of 50% (45% for C4 plants) is recommended. Therefore, we now use these values and have recalculated the simulations accordingly.** Since the literature information for this threshold value is still rather scarce, we have additionally checked the impact of the threshold parameter by varying it from 30 to 70%. The sensitivity analysis showed that the change in biomass production was between 0 and 11 percent depending on sites and crops. By using 50% for all investigated crops the results vary by +/- 7 percent at a maximum. This indicates that the parameter is not overly sensitive and that the selected values are a reasonable choice.
For applying the model at less known sites (not in this investigation), we regard the total extractable water as directly related to the soil type (e.g. Sand, Clay, Loam, etc.) and thus only indirectly to vegetation type (since agricultural vegetation is generally grown on more fertile soils).

R2: For me, the LAI at Niakhar was underestimated, and that at Wankama 1 was overestimated, NEE at Bontioli was overestimated, is it possible to have more discussion about the reason or adjust the parameters to have a better capture?

A2: Thank you for the remark. Indeed, the local measurements of LAI in Niakhar indicate a slightly higher LAI than resulted from the model while the remote sensing misses the peaks that we represent in the Savanna region of Wankama 1. Regarding Niakhar, trees in the area that are not considered for the simulation of agricultural site likely lead to a higher LAI than simulated. In Wankama, this is likely related to the widespread occurrence of bushes in the region that were not considered in the modelled grassland type, leading the NDVI and thus LAI estimates to be larger outside the season but providing competition to the grass which prevents its full biomass development. **We have now included these discussion points into the text supplying also literature references that support these arguments (L360ff, L431ff).**

R3: There is a long description about the model in 2.3.1, I suggest adding the important equations to make it easy to understand the revenant biophysical process of the model since not all the readers are familiar with the model. For example, how is the actual evaporation calculated from the potential evaporation? Also, the Thornthwaite approach mainly depends on temperature, it seems water content is important in this special region with large variation of precipitation, so, how is the impact on the result? How is the performance of the modelled evaporation compared to the flux observation?

A3: Since LDNDC is a rather complex model, it is difficult to supply a few equations that are most important. In particular if the description should not be extended. **However, we have gone through the description in order to clarify potential difficulties. In particular, we have now better explained the Thornthwaite approach, which indeed is based on temperature (and daylength period), and added the description of how potential evapotranspiration is used to derive potential transpiration demand as well as the various evaporation fluxes (L255ff).**
The performance of the model to represent water fluxes above grassland and groundnut has been evaluated at the Bontioli site in a previous publication (Grote et al. 2009b). Regarding the other sites, we compared evaporation data from graphs in various publications of the sites Agoufou 2006-7, Wankama 2005-6, and Sumbrungu/Nazinga/Kayoro 2013 which were visually in good accordance to the simulated fluxes. **In order to demonstrate that the water balance is reasonably well represented, we have introduced the measured soil water conditions at each site along with the simulated ones (new Fig. 3, 5, and 7). The comparison of both shows that the dynamics are very well met, with minor deviations at Nalohou and Agoufou only, where the water storage capacity has been underestimated in both cases. Results are additionally discussed (L366ff, L412ff).**

R4: How is the parameters relevant to soil water content (field capacity and wilting point) at each site? Please clarify.

A4: In order to save space, **we added the water holding capacity (total field capacity – wilting point in mm m-3) for each site into Table 1.**

R5: I suggest adding the vegetation distribution in Fig. 1, which is more intuitive and easier to understand.

A5: Thanks for the recommendation. **We added the vegetation type distribution into Fig. 1 accordingly.**

R6: The authors used the Modis LAI, which is 500-m resolution, if the grid is a mixture of different vegetation, it might have a big impact on the validation. Such discussions are needed.

A6: We agree that this issue is worth investigating in more detail. **Therefore, we have added a comparison of MODIS derived and site-specific LAI measurements (new Fig. 2) and discuss the deviations between both (L205ff). Indeed, the resolution of MODIS data is an important point here, which is further corroborated by new literature cited.**

R7: Line 318-320, how is the standard for spinup? It says it accounts for the competition on light and water at the sites. Please clarify this in detail.

A7: The standard spin-up procedure has been previously explained in the same chapter with regard to agricultural sites. It consists of a three-year simulation run that is primarily needed to account for uncertain initializations of soil carbon and nitrogen pools. In case of grasslands, where the biomass initialization is also highly uncertain and varies from year to year, this spin-up period also helps to avoid steep adjustments due to specific site conditions during the first years. **Since we recognized that this description should not be placed into different paragraphs, we now join the text passages and put it in front of chapter 2.3.2 (Model setup, L278ff).**

R8: Minor, I think gas exchange might be removed from the title since it is not reflected in the main body.

A8: We beg your pardon but after some consideration we think that the relatively extensive NEE modelling part justifies mentioning gas exchange in the title.

R9: Minor, I fell the abstract is quite long and should be more concise.

A9: Thank you for the recommendation. **We have gone through the abstract and made it more stringent.**

R10: Minor, superscript in the figure is ignored, please modify.

A10: Thank you for noticing. **We corrected the letter style in the Figures (now 4, 6, 8).**

Reviewer 2:

R1: Lines 203-207: Does the author validate the MODIS LAI data with field measure LAI in the studied sites?

A1: This is a bit tricky since field observations of LAI were only occasionally done within the period from when the MODIS data were derived. Nevertheless, it is an interesting task and thus we added a comparison using only maximum LAI values at a site per year (new Figure 2). Generally, the observations match the MODIS data but are higher at higher LAI levels. **We add a brief description, possible explanations and additional discussions (including literature references, L211ff).**

R2: Lines 209: I would put some key functions, such as the calculation of water stress factor, in this section.

A2: After some consideration, we have decided against this suggestion, since it is difficult to supply the most important equations in a rather complex model, in particular since the importance might be rather different for different vegetation types. **However, we went through the description and clarified potentially difficult issues such as the calculation of evapotranspiration (L253ff) which also led to the indication of new parameters in tables 2 and 3.**

R3: Some results were not fully discussed. For example, in Figure 1(A), the LAI was overestimated in 2007 and the NEE was overestimated at the very beginning of the growing season.

A3: Thank you for the remark. Indeed, the local measurements of LAI and NEE were not always well represented. Several reasons might be considered but the most important relates to uncertainties in the vegetation initialization (e.g. at Niakhar trees were likely considered in the MODIS pixel leading to an overestimation of LAI. Similar, in Wankama, bushes are widespread in the region but not considered in the modelled grassland type). **We have now improved the discussion and included the specific points mentioned to the text (L360ff, L431ff, L500ff).**

R4: Although this manuscript focus' on evaluating the ability of LandscapeDNDC in simulating gas exchange and biomass production simulation, the simulation of soil moisture is very important especially in some arid or semi-arid sites (such as Agoufou, Dahara, and Wankama). Do you have the field measured soil moisture? If you have, I suggest the author validate the simulation of soil moisture of LandscapeDNDC.

A4: Thank you for this suggestion. The performance of the model to represent water fluxes above grassland and some crop sites has been evaluated for Bontioli in a previous publication (Grote et al. 2009). **However, we have now provided additional soil water content measurements for all sites in Fig. 3 and 5 for additional evaluation.**

R5: For the validation of NEE, I would like to partition NEE into gross primary productivity and ecosystem respiration. After that, the gas exchange part can be in-depth discussed.

A5: This is true and in fact, simulated NEE is a composite from the model outputs of gross primary production and respiration. Similar comparisons have been done e.g. in Lindauer et al. and Molina-Herrera et al. before. However, the paper is already quite long and a discussion that would need to include the uncertainties related to the gas partitioning processes of the LDNDC model as well as the empirical algorithm used to separate NEE into two fluxes would substantially increase the extent of the manuscript. We also think that the net flux which is the only one directly measured is also the most important and is corroborated by the parallel development of biomass.

Lindauer, M., Schmid, H. P., Grote, R., Mauder, M., Steinbrecher, R., and Wolpert, B.: Net ecosystem exchange over a non-cleared wind-throw-disturbed upland spruce forest – measurements and simulations Agric. Forest Meteorol., 197, 219-234, 10.1016/j.agrformet.2014.07.005, 2014.

Molina-Herrera, S., Grote, R., Santabárbara-Ruiz, I., Kraus, D., Klatt, S., Haas, E., Kiese, R., and Butterbach-Bahl, K.: Simulation of $CO_2$ fluxes at European forest ecosystems with the coupled soil-vegetation process model "LandscapeDNDC", Forests, 6, 1779-1809, 10.3390/f6061779, 2015.

---

## Referee Report (RR1)

The authors have addressed my previous comments. I only have few minor comments to help further improve the manuscript.

R1: The authors underscored the importance of the soil water availability to this special region and ecosystem, they considered 70% of the extractable soil water content, I am wondering whether it is vegetation specific since there are three distinct types here (croplands, pure grasslands as well as tree/grass mixtures)? I suggest adding a sensitivity test to illustrate its influence. Or, just set different thresholds for different vegetation (this might be doable since the simulation here was conducted at site level instead of regional level).

A1: The extractable soil water has in fact been set from field measurements per site as the difference between field capacity and soil wilting point, stratified in different layers. The 70% of extractable water has been set as the threshold value after which photosynthesis is linear reduced until it is zero when the wilting point is reached. **For consistency reasons we thus added the respective parameter H2OREF_A into Table 3 and refer to it in the text. However, we found in the FAO database that 70% is considered rather at the upper end of the range and a value of 50% (45% for C4 plants) is recommended. Therefore, we now use these values and have recalculated the simulations accordingly.** Since the literature information for this threshold value is still rather scarce, we have additionally checked the impact of the threshold parameter by varying it from 30 to 70%. The sensitivity analysis showed that the change in biomass production was between 0 and 11 percent depending on sites and crops. By using 50% for all investigated crops the results vary by +/- 7 percent at a maximum. This indicates that the parameter is not overly sensitive and that the selected values are a reasonable choice.

For applying the model at less known sites (not in this investigation), we regard the total extractable water as directly related to the soil type (e.g. Sand, Clay, Loam, etc.) and thus only indirectly to vegetation type (since agricultural vegetation is generally grown on more fertile soils).

Ccomment: The authors have considered FAO values, this is acceptable. The authors mentioned the sensitivity analysis in this reply, but without showing specific figures or tables. So, I suggest adding them (in the supplementary is also fine) to have a better understanding on the uncertainties.

R3: There is a long description about the model in 2.3.1, I suggest adding the important equations to make it easy to understand the revenant biophysical process of the model since not all the readers are familiar with the model. For example, how is the actual evaporation calculated from the potential evaporation? Also, the Thornthwaite approach mainly depends on temperature, it seems water content is important in this special region with large variation of precipitation, so, how is the impact on the result? How is the performance of the modelled evaporation compared to the flux observation?

A3: Since LDNDC is a rather complex model, it is difficult to supply a few equations that are most important. In particular if the description should not be extended. **However, we have gone through the description in order to clarify potential difficulties. In particular, we have now better explained the Thornthwaite approach, which indeed is based on temperature (and daylength period), and added the description of how potential evapotranspiration is used to derive potential transpiration demand as well as the various evaporation fluxes (L255ff).**
The performance of the model to represent water fluxes above grassland and groundnut has been evaluated at the Bontioli site in a previous publication (Grote et al. 2009b). Regarding the other sites, we compared evaporation data from graphs in various publications of the sites Agoufou 2006-7, Wankama 2005-6, and Sumbrungu/Nazinga/Kayoro 2013 which were visually in good accordance

to the simulated fluxes. **In order to demonstrate that the water balance is reasonably well represented, we have introduced the measured soil water conditions at each site along with the simulated ones (new Fig. 3, 5, and 7). The comparison of both shows that the dynamics are very well met, with minor deviations at Nalohou and Agoufou only, where the water storage capacity has been underestimated in both cases. Results are additionally discussed (L366ff, L412ff).**

Comment: Thanks for adding the comparison of soil water content, is the simulation for 10 cm? please clarify the methods sections on the soil water content simulation. How about the average root depth for each vegetation/crop at each site, why choose 10-cm soil water content to compare?

I cannot totally agree with the authors that it is only minor deviations at the Nalohou and Agoufou site. It seems a serious underestimation especially at Nalohou site. I am wondering is it attributed to the overestimation of the evapotranspiration? I recommend more discussion for this relatively large bias.

R7: Line 318-320, how is the standard for spinup? It says it accounts for the competition on light and water at the sites. Please clarify this in detail.

A7: The standard spin-up procedure has been previously explained in the same chapter with regard to agricultural sites. It consists of a three-year simulation run that is primarily needed to account for uncertain initializations of soil carbon and nitrogen pools. In case of grasslands, where the biomass initialization is also highly uncertain and varies from year to year, this spin-up period also helps to avoid steep adjustments due to specific site conditions during the first years. **Since we recognized that this description should not be placed into different paragraphs, we now join the text passages and put it in front of chapter 2.3.2 (Model setup, L278ff).**

Comment: How about the field management at each site? Is there no irrigation and fertilization as the simulation settings?

Minor comment:

Line 259, please define the RWC at the first appearance here?

---

## Author Response (AR2)

**Point-by-point response for:**

gmd-2020-417: 'Modelling Gas Exchange and Biomass Production in West African Sahelian and Sudanian Ecological Zones'

**General remarks**

We are grateful for the general positive evaluation of the manuscript and have now addressed the remaining issues and questions that will certainly improve the comprehensiveness of the text.

Specific answers to the remarks (R: Repeated remark of the reviewer; A: Answer, **concrete changes in the text as indicated with bold letters**)

**Reviewer 1**

Comment to R1: The authors have considered FAO values, this is acceptable. The authors mentioned the sensitivity analysis in this reply, but without showing specific figures or tables. So, I suggest adding them (in the supplementary is also fine) to have a better understanding on the uncertainties.

A Reply to CR1: it is true that we have only indicated the parameter values and the FAO reference but do not report about the sensitivity analysis. Since we don't think to supply an extra table or figure on this or build a supplement file particular for this issue, we decided to supply this information within an additional paragraph added to table 3, with a reference link to the parameters **(L335ff)**.

> **"Since there is a certain variability for this parameter (H2OREF_A) with respect to species and sites, we conducted an additional analysis to assess how sensitive biomass production of each crop species responds to this parameter. To do this, we varied H2OREF_A between 0.30 to 0.70 at each site and for each species. Results indicated that when applying the upper limit of 0.7, productivity are lower than the standard value for C3 (0.5) and C4 (0.45) crops by 6.4% for peanut, 3.9% for millet, 1.9% for cassava, 0.78% for sorghum, and 0.11% for maize. On the other hand, applying the lower limit of 0.3 increased productivity relative to the standard value by 3.8% for peanut, 1.5% for millet, 1.1% for cassava, 0.07% for sorghum, and 0.05% for maize. Thus, the overall sensitivity of biomass production to the RWC threshold value of photosynthesis decline was judged to be low."**

Comment to R3: Thanks for adding the comparison of soil water content, is the simulation for 10 cm? please clarify the methods sections on the soil water content simulation. How about the average root depth for each vegetation/crop at each site, why choose 10-cm soil water content to compare?

I cannot totally agree with the authors that it is only minor deviations at the Nalohou and Agoufou site. It seems a serious underestimation especially at Nalohou site. I am wondering is it attributed to the overestimation of the evapotranspiration? I recommend more discussion for this relatively large bias.

Reply to CR3: The water content down to 10cm depth has been used because these data were available at all sites and showed the least gaps, enabling the best comparison of performances between sites. In addition, a depth close to the surface is more responsive to rainfall occurrences and is also most important ecologically since generally root abundance is higher than in deeper soil layers. We added an description and the limitation considerations into the methodology sector **(L176ff).**

Regarding the deviations between measured and simulated soil water content at the sites Nalohou and Agoufou, we tested some further model settings and concluded that they are mostly related to the soil parameterization, i.e. the field capacity and wilting point. For example, lowering the wilting point at Agoufou by

approximately 20 mm, would considerably improve the match with measured SWC compared to using the reported soil properties. This uncertainty in the initialization of soil properties that is difficult to derive with certainty in a heterogeneous footprint area is now discussed **(L388ff, L429ff)**.

Comment to R7: How about the field management at each site? Is there no irrigation and fertilization as the simulation settings?

Reply to CR7: No, there is indeed no fertilization and irrigation officially reported for the sites, so we don't consider any in the simulations too (see also the documented input files in the data repository filed under https://radar.kit.edu/radar/en/dataset/LpgXAmcqzUCGPdga?token=YbkhsPYRBKZklhsrxOdV. This is now explicitly mentioned in the methodology sector **(L289).**

Minor comment: Line 259, please define the RWC at the first appearance here?

Reply to MC: It is true, the first appearance of RWC is at L259 (**now L264**), not L275. This has been corrected.